# Changes of Microbial Diversity in Rhizosphere Soils of New Quality Varieties of Winter Wheat Cultivation in Organic Farming

**Anna Gałązka [1],\*** , **Emilia Grzęda [1]** and **Krzysztof Jończyk [2]**

1    Department of Agriculture Microbiology, Institute of Soil Science and Plant Cultivation—State Research Institute, Czartoryskich 8, 24-100 Pulawy, Poland

2    Department of Systems and Economics Crop Production, Institute of Soil Science and Plant Cultivation—State Research Institute, Czartoryskich 8, 24-100 Pulawy, Poland

\*    Correspondence: agalazka@iung.pulawy.pl; Tel.: +48-814-786-950

**Abstract:** The aim of this paper was to evaluation functional diversity in rhizosphere soils of new quality varieties of winter wheat cultivation in organic farming. Field experiments were carried out in 2017 and 2018. Twelve commercial winter wheat varieties were selected for testing: Arktis, Bellisa, Estivus, Fidelius, Hondia, Jantarka, KWS Ozon, Linus, Markiza, Ostka Strzelecka, Pokusa, and Rokosz. Winter wheat cultivars were chosen for their high yielding potential and good tolerance to fungal diseases. In the plant production conducted in accordance with the principles of organic farming, the selection of the best quality varieties is a key element of agrotechnics. The samples of rhizosphere soils were collected each year in two seasons: spring and summer. The basic parameters of soil biological activities and microbial biodiversity indicators were determined. The high variability of biological activity and functional diversity of rhizosphere soils in the growing season between particular varieties of winter wheat was observed. The rhizosphere soils from varieties such as Bellisa, Arktis, Jantarka, Fidelius, Ostka Strzelecka, Pokusa, Rokosz and KWS Ozon were characterized by high biological activity and functional biodiversity. On the other hand, the soils collected from the varieties Estivus, Fidelius, Jantarkaand Hondia were characterized by medium and low biological activity and biodiversity indices. The highest yield was found in winter wheat varieties such as Bellisa, Fidelius and Jantarka. The results of these analyses allows for a more complete characterization of the yield potential of the tested varieties and their suitability for cultivation in the conditions of organic farming, taking into account the biological activity of soils.

**Keywords:** ecological system; winter wheat; cultivars; biological activity; microbial diversity

## 1. Introduction

The quality of the soil environment is inextricably linked to soil cultivation methods. Intensive farming results in substantial degradation of the soil environment, which forces a constant search for techniques supporting the protection of soil functions and biodiversity [1,2] Sustainable agriculture (including organic farming), whose assumptions are focused on the preservation of the natural environment and an increase in production without excessive interference with the natural environment resources, is based on the support of natural biological processes without disturbing processes reproducing the biocenosis life and the natural soil structure [3,4].

In recent years organic farming has been implemented quite intensively in many countries. The assumptions of this farming do not involve the use of hazardous chemicals and are the basis for obtaining high quality crops [5,6]. Cereals cultivation is one of the main trends in organic production.

The selection of the best quality varieties is a key element of agrotechnics particularly in plant production conducted in accordance with the principles of organic farming [7,8]. There is the possibility of selecting the high-quality varieties of cereals for a given field. The biological activity and health of the soil on which the cultivation is carried out is also of great importance here. In the case of some cereal varieties, microorganisms more easily establish symbioses and develop more actively in the soil rhizosphere [9]. This is mainly related to the genotype of the plant [7]. Selecting the right variety allows a better use of the potential of the habitat, preventing pathogens and creating high-quality crops. Therefore, the selection of new varieties is of key importance in connection with the biodiversity and biological activity of soils [10,11]. Both in Europe and abroad, there is a growing interest in biological methods of increasing the yield and quality of crops, caused both by an increased interest in sustainable and organic farming, as well as by the agricultural producers' clear economic benefits resulting from the use of properly selected cereal varieties [12,13].

In many countries, mainly due to the small area of cultivation, there is no special plant breeding for the needs of organic farming. In this case, cultivation in organic farms is recommended to choose varieties that are in the general offer of breeding companies and located in the National Register of Varieties. Research conducted by breeding crop varieties centers and recommendations of breeding companies do not take into account the assessment of varieties in the conditions of organic farming, which hampers proper selection and increases the risk of cultivation [14]. In addition, there is no study of the biological activity and functional diversity of soils in this respect depending on the variety of cereals used. In the rhizosphere of plants, an increased activity of microorganisms is observed, which contributes to the more effective activation of minerals contained in the root soil, and thus to better nutrition of plants [15,16]. In previous studies on rhizosphere microorganisms, the focus was on determining their number and identification of the most important species. It is now also possible to determine the metabolic profile of soil microorganisms and to assess their functional biodiversity [17,18] (Gałązka et al., 2017).

Soil fertility and biological activity can be increased by using appropriate crop rotations, fertilization, selection of plant and proper water and air relations [19]. In order to protect the natural environment and to restore natural soil fertility, solutions are increasingly sought to limit chemization and restore soil biodiversity [8]. The number and activity of microorganisms is determined by many biotic and abiotic factors. However, the main factor limiting their development is the availability of organic matter [20]. The composition of microorganisms can be an important determinant of the rate of organic matter distribution and circulation of nutrients and their availability in soils. The use of selected plant varieties and related bacterial consortia in ecological crops should significantly increase both the biological activity of soils and their fertility in organic farming. Such applications will be the basis for the development of innovative technology for increasing the growth and yield of cereal plants.

The concept of the presented study includes the assessment of the suitability for cultivation of the latest quality varieties of winter wheat conducted in the conditions of organic farms. The results obtained constitute part of the assumptions of the varietal assessment system for the needs of organic farming—Organic Verification Experimentation.

The aim of the study was the evaluation of changes of microbial functional diversity in rhizosphere soils of twelve new quality varieties of winter wheat cultivation in organic farming. The aim of the paper also includes indication of agricultural and physiological characteristics relevant for organic forming, which are not taken into account in the standard assessment of varieties, i.e., biological activity and biodiversity of soils.

## 2. Materials and Methods

### 2.1. Field Experiment

The study was based on the stationary field experiment established on a Haplic Luvisol (loamy sand, according to USDA) in Osiny Experimental Station (51°28′ N, 22°30′ E), Poland. The basic properties

of soil are as follows: C$_{org.}$—8.2 g/kg; N$_{tot.}$—1.0 g/kg; pH$_{KCl}$—5.8; P, K and Mg (mg/kg$^{-1}$)—37.5, 84.7 and 54.5, respectively. The following crops are rotated: potato, spring wheat (under-sown with clover-grass mixture), clover-grass mixture, winter wheat, oats-field peas mixture. In 2017 and 2018 twelve commercial winter wheat cultivars were grown: Arktis, Bellisa, Estivus, Fidelius, Hondia, Jantarka, KWS Ozon, Linus, Markiza, Ostka Strzelecka, Pokusa, and Rokosz. In the absence of winter wheat varieties for the purpose of organic farming, cultivars were chosen for their high yielding potential and good tolerance to foliar fungal diseases. Each cultivar was sown (4.5 million grains/ha) in 3 replicated and randomized micro-plots (35 m$^2$). No nitrogen mineral fertilizers were used in the organic system, but regarding phosphorus, 150 kg/ha of powdered phosphate rock is applied once per rotation. With respect to K fertilization, the winter wheat field was treated with 150 kg/ha of potassium sulfate before sowing. Organic fertilization in this system included 30 Mg/ha of compost plowed in before potato planting. No synthetic pesticides were applied to control pests, and spring harrowing was used to reduce weed infestation in winter wheat plots.

## 2.2. Soil Samples Collection

Soil samples were collected in spring and summer (2017–2018) from the rhizosphere zone of plants, pooled in individual plastic bag and transferred to portable refrigerator. From each micro-plots (layer of 0–20 cm) were taken several subsamples and combined from one (biological replicate). In this way, 3 biological repetitions were obtained. Fresh samples were sieved through a 2 mm sieve, collected in sterile, plastic tubes, and stored at 4 °C until biological activities measurements. This ensures that the data collected by using different methods (enzymatic activity, microbial diversity, substrates utilization, etc.) depends only on the winter wheat varieties and their characteristics.

## 2.3. Bacterial Communities and Enzymatic Analysis

Microbiological counts were expressed as a number of colony forming units (CFUs) per g of dry soil. Total number of microorganisms was determined by the dilution method on agarized soil extract. The total number of fungi was determined on Martin's medium [21]. The enzymatic activities were determined spectrophotometrically: soil dehydrogenase activity using TTC method [22], and phosphatases activity by *p*-NPP method [23]. The activities of acid (ACP) and alkaline phosphatase (ALP) were analyzed using 1 g of soil incubated for 1 h (37 °C) at their optimal pH (pH = 11 alkaline; pH = 6.5 acidic) with p-nitrophenyl phosphate. The soil dehydrogenase activity (DHA) was determined in 6 g soil by colorimetric measurement of the reduction of 2,3,5-triphenyltetrazolium chloride (TTC) solution, after incubation at 37 °C for 24 h. The enzyme activity was measured spectrophotometrically at 485 nm.

## 2.4. Community Level Physiological Profiling Technique (CLPP)—Biolog EcoPlate Methods

Firstly, 1 g of fresh soil was taken and suspended in a bottle containing 99 mL of sterile water, followed by shaking for 20 min at 20 °C. The suspension was then left to settle for 20 min at 4 °C. Supernatant was filtered through BagFilter® 400P to avoid transmission of the remaining soil particles. Inoculation was accomplished by pipetting 120 μL of samples to each well of the Biolog EcoPlate. Plates were incubated at 25 °C for 168 h and the 590 nm absorbance was measured every 24 h. The average well color development (AWCD) was calculated for each group of substrates, assuming OD$_{590}$ = 0.25 as a threshold value, below which a substrate is considered as unmetabolized by the method described by Garland [24]. A full description of the methodology was described in a paper by Gałązka and Grządziel [25].

## 2.5. Statistical Analyses

Statistical analyses were performed using the packet STATISTICA.PL (10) (Stat. Soft. Inc., Tulsa, OK, USA). The collected data were subjected to analysis of variance (ANOVA) for the comparison of means, and significant differences were calculated according to post-hoc Tukey's HSD test at *p* < 0.05

significant level. The cluster analyses and HeatMaps were performed on standardized data from the average absorbance values at 144 h (Biolog FFPlate). The results were also submitted to the PC (principal component) analysis to determine the correlations between the varieties of winter wheat cultivation, season, year and biological activity.

## 3. Results

### 3.1. Meteorological Conditions during the Experiment

The evaluation of the soil biological activity in the rhizosphere of different varieties of winter wheat cultivated in organic farming was carried out on two groups of analyses: the classic analyses of soil biological activity, including the total number of bacteria, fungi and enzymatic activity, and analyses of metabolic profile of the microorganism using Biolog EcoPlates.

The meteorological conditions during the growing season of twelve varieties of winter wheat cultivation in organic farming are presented in Table 1. The presented data show that both in 2017 and in 2018 the weather conditions were similar, which makes it possible to compare the results obtained in the different years. Soil samples were collected each year in May (spring) and July (summer). In addition, samples were taken every year under the same soil moisture conditions. Both the sum of precipitation and the temperature in July 2017 and 2018 were the same level.

**Table 1.** Meteorological conditions during the growing seasons (2017–2018).

| Month | Year | | Average from Years 1871–2000 |
|---|---|---|---|
| | 2017 | 2018 | |
| Sum of precipitation (mm) | | | |
| III | 32 | 31 | 28 |
| IV | 65 | 30 | 42 |
| V | 62 | 59 | 55 |
| VI | 31 | 38 | 71 |
| VII | 109 | 122 | 78 |
| VIII | 96 | 28 | 67 |
| Average air temperature (°C) | | | |
| III | 6.0 | 0.4 | 1.9 |
| IV | 7.6 | 13.6 | 8.1 |
| V | 13.6 | 17.2 | 13.8 |
| VI | 18.1 | 18.8 | 17.1 |
| VII | 18.6 | 20.7 | 18.6 |
| VIII | 19.6 | 20.7 | 17.6 |

### 3.2. Microbial Diversity and Enzymatic Activities

Based on a multivariate ANOVA analysis, it was found that the winter wheat variety, the season and the year of sampling were significant in assessing the soil's biological activity (Table 2). As variables in the analysis, the following activities were chosen: enzymatic activity, total number of bacteria, total number of fungi and indicators of soil functional metabolism evaluation on the example of Biolog EcoPlate analysis. Detailed information is presented in Table 2. In the further part of the work, the analysis results will be discussed, taking into account three discussed factors (variety, season, year).

**Table 2.** Results of a multidimensional analysis of variance ($p \leq 0.05$) for the entire data set. Multidimensional significance tests. Parameterization with sigma-restrictions-decomposition of effective hypotheses.

| Analysis of Variance | Wilks Value | F Value | $p$ |
|---|---|---|---|
| Varieties | 0.000028 | 657,185.0 | 0.000000 |
| Year | 0.000000 | 199.1 | 0.000000 |
| Season | 0.680755 | 8.6 | 0.000001 |
| Varieties_Year | 0.004022 | 4556.7 | 0.000000 |
| Varieties_Season | 0.040512 | 7.8 | 0.000000 |
| Year_Season | 0.000002 | 129.8 | 0.000000 |
| Varieties_Year_Season | 0.674009 | 8.9 | 0.000001 |

In 2017–2018 the highest total bacteria number was characterized by the rhizosphere soil of three wheat varieties collected in spring such as Pokusa, KWS Ozon and Linus and was respectively $767.2 \times 10^7, 457.4 \times 10^7$ and $332.8 \times 10^7$ CFU/g d. m. of soil (Figure 1). The medium total bacteria number was characterized by varieties of Estivus, Arktis and Fidelius (from 265.8 to $232.6 \times 10^7$ CFU/g d.m. of soil). The remaining varieties of winter wheats in samples collected in spring were characterized by relatively low populations of the rhizosphere microorganisms (Figure 1). The lowest total bacteria number in spring samples was characterized by the rhizosphere soil of the Hondia variety.

In addition, significantly higher overall bacterial counts were found in rhizosphere soil in the spring compared to the summer (Figure 1). A higher total bacteria number was observed, among others in the rhizosphere soil of the varieties Pokusa, KWS Ozon, Linus and Arktis in the second phase of sampling (summer). In the summer the highest total bacteria number was found in the rhizosphere of wheat varieties, i.e., Pokusa ($283.7 \times 10^7$ CFU/g d. m. of soil), Estivus ($224.4 \times 10^7$), Fidelius ($211.4 \times 10^7$), Linus ($193.1 \times 10^7$) and Jantarka ($174.7 \times 10^7$) (Figure 1).

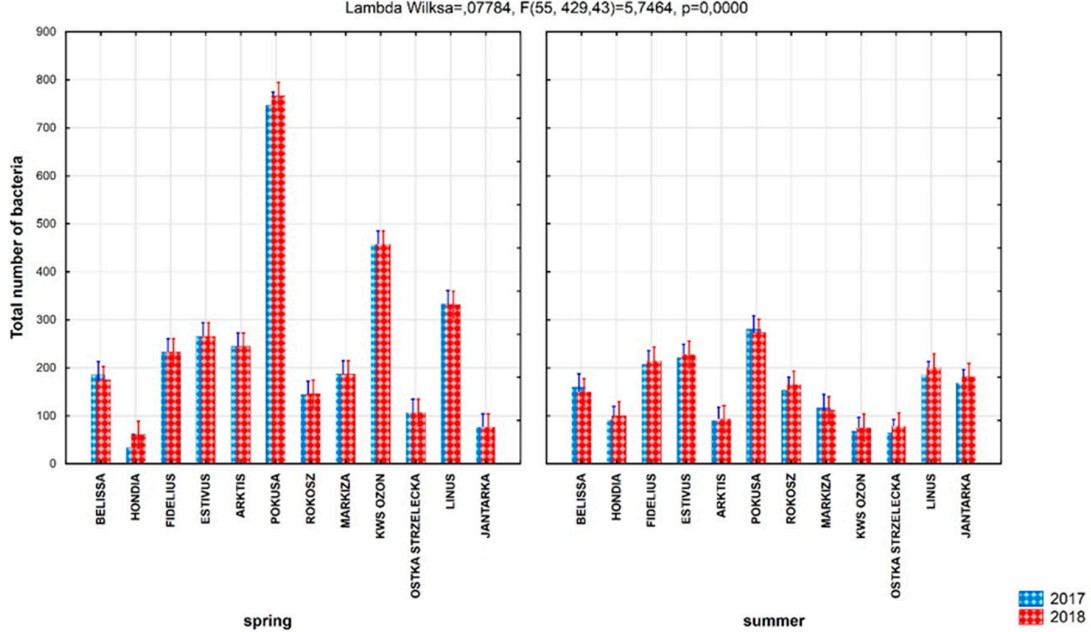

**Figure 1.** Total number of bacteria ($10^7$ CFU/g d. m. of soil)—three-way analysis of variance.

The dynamics of soil microorganism development expressed by changes in the number of the particular groups of bacteria and fungi settled in this environment, i.e., the total number of bacteria, fungi and many others, is a measurable indicator of the biological life of soil.

Similarly to bacteria, soil fungi in the first stage of sampling (spring) also occurred most frequently in the rhizosphere of the Arktis variety ($32.0 \times 10^4$ CFU/g d. m. of soil), KWS Ozon ($27.8 \times 10^4$) and Linus ($27.0 \times 10^4$) (Figure 2). However, such varieties as Jantarka and Hondia were characterized by significantly lower populations of fungi, from $14.4 \times 10^4$ to $16.7 \times 10^4$ CFU/g d. m. of soil for the variety, respectively. The highest number of fungi in samples collected in the summer was found in the soil of the winter wheat variety Estivus ($33.8 \times 10^4$ CFU/g d. m. of soil). In addition, there was a significantly higher total fungi number in the rhizosphere soil in the first phase (spring) of soil samples collection as compared to the second phase (summer) (Figure 2).

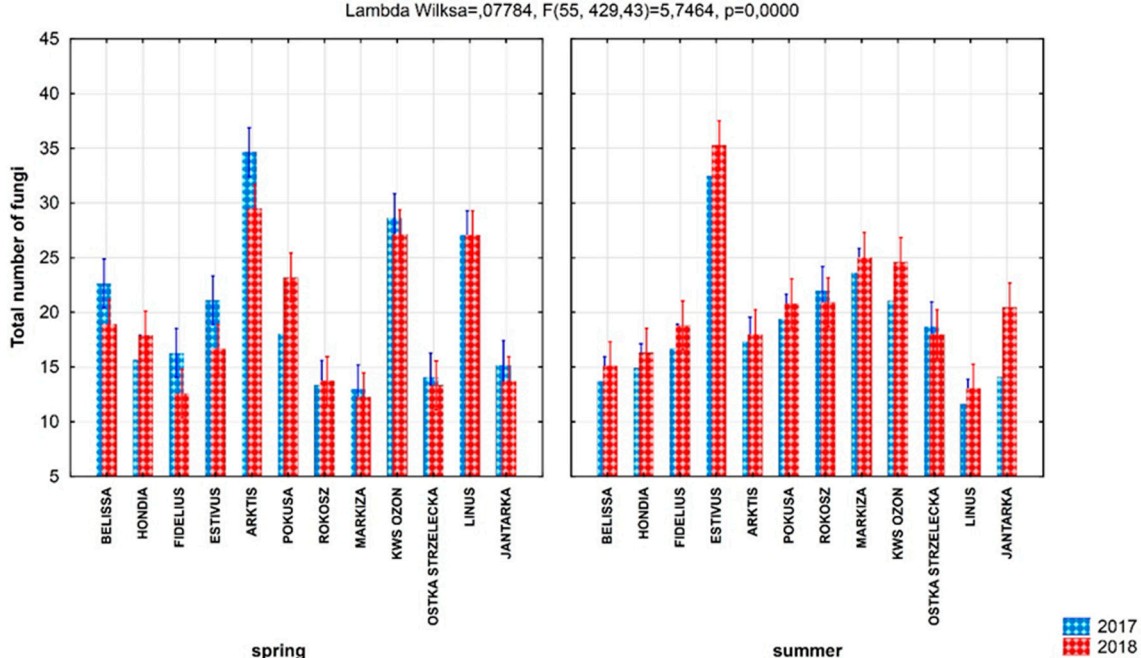

**Figure 2.** Total number of fungi ($10^4$ CFU/g d. m. of soil)—three-way analysis of variance.

Dehydrogenases, as respiratory chain enzymes, play a major role in energy production by organisms. The highest dehydrogenase activity was observed in the rhizosphere soil of the following wheat varieties: Fidelius, Estiwus, Arktis, Pokusa, Rokosz collected in spring, and Rokosz for the second phase of sampling, i.e., summer (Figure 3). The rhizosphere soil from the Rokosz cultivar in both seasons was the highest activity of dehydrogenases. High dehydrogenase activity is indicative of the high metabolic activity of microorganisms inhabiting the rhizosphere soils. Significant correlation coefficients between the number of analyzed groups of microorganisms, (i.e., bacteria and fungi), and the activity of dehydrogenases, respectively: r = 0.831 and r = 0.845, indicated close relationships between these variables.

Soil phosphatase can also be a good indicator of biological activity in soil. Determining alkaline and acid phosphatase in the soil samples gives us large amount of information about biological characteristics of the soil. Phosphatase activity in soil reflects the activity of enzymes associated with soil colloids and humic substances and frees phosphatases in the soil solution and phosphatases associated with live and dead cells, plants, and microorganisms. In the case of acid phosphatase activity (Figure 4), the highest activity of this enzyme was found in the rhizosphere soil of the Estivus variety in soil collected from rhizosphere soil in summer (Figure 4). In the soils collected in spring, the highest activity of acid phosphatase was characterized by soil from wheat varieties Belissa, Estivus and Arktis. In summer samples, high activity of acid phosphatases was also observed in the following cultivars: Pokusa, Rokosz, Markiza, Ostka Strzelecka and Linus (Figure 4).

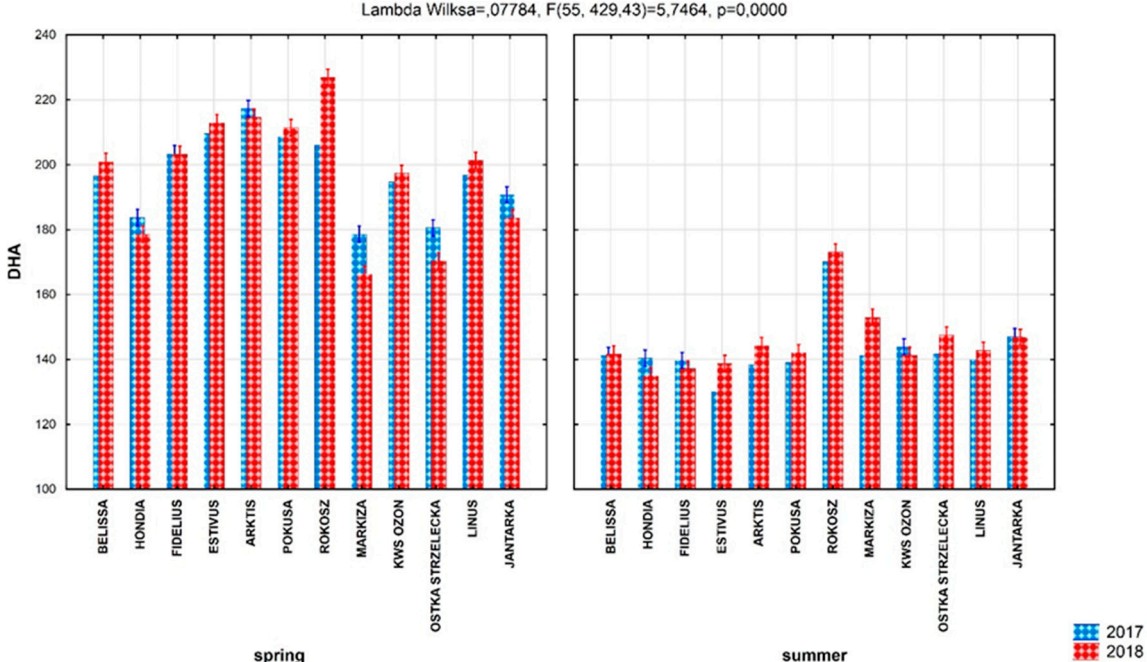

**Figure 3.** The dehydrogenases activity (DHA) (ug formazan/g d. m. of soil/24 h)—three-way analysis of variance.

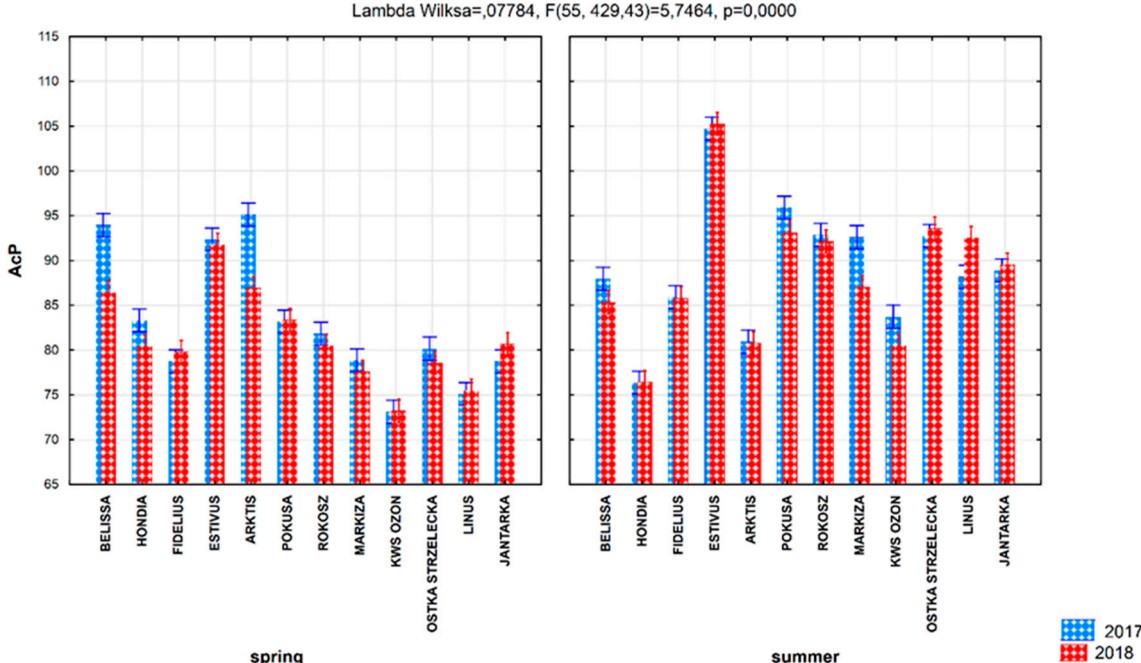

**Figure 4.** The acid phosphatase (AcP) (ug p-nitrophenol/g d. m. of soil/h)—three-way analysis of variance.

On the other hand, the highest alkaline phosphatase activity was found in the rhizosphere soil collected in spring of KWS Ozon cultivar, but higher alkaline phosphatase activity was also observed in this season in soil under varieties such as Hondia, Estivus and Arktis (Figure 5). In soil collected from rhizosphere soil in summer the higher alkaline phosphatase activity was found in the rhizosphere soil such varieties as Hondia, KWS Ozon and Jantarka. Additionally, the higher alkaline phosphatase activity was observed in soil collected from varieties Belissa, Fidelis, Markiza and Linus.

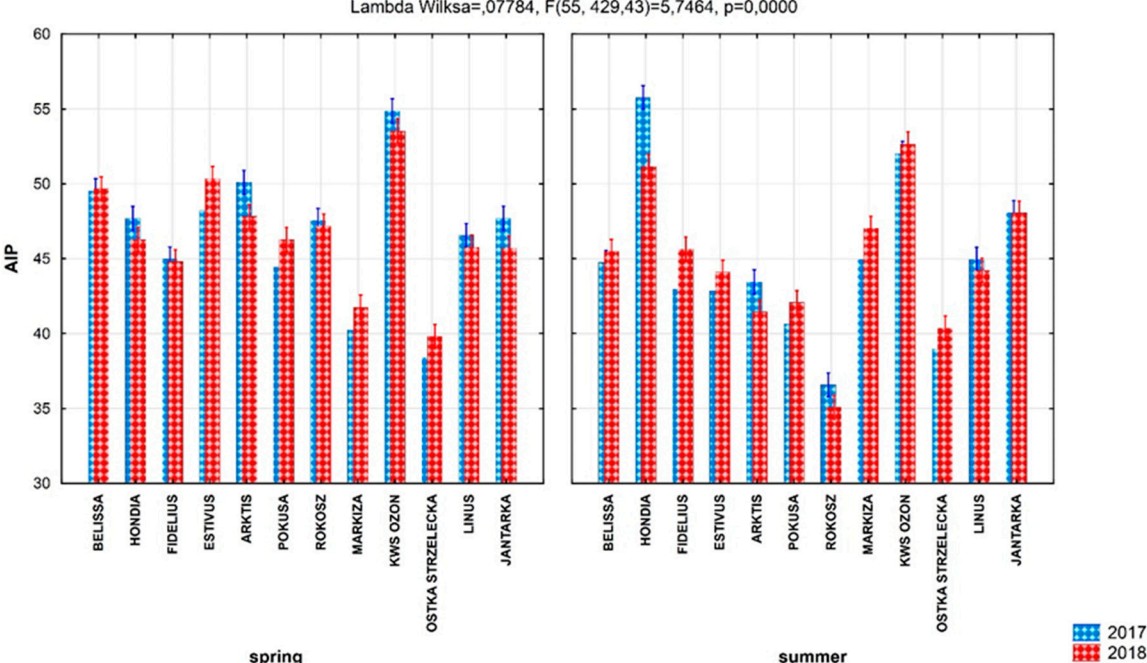

**Figure 5.** The alkline phosphatase (AlP) (ug p-nitrophenol/g d. m. of soil/h)—three-way analysis of variance.

### 3.3. Functional Diversity of Soil—Community Level Physiological Profiles (CLPP)

Based on the Biolog EcoPlates analysis, the biodiversity indicators and 31 carbon source utilization for rhizosphere soils of twelve winter wheat varieties were calculated.

Average well-color development ($AWCD_{590}$) for each soil sample was calculated based on the absorbance for 31 substrates, less the absorbance for pure water. $AWCD = \Sigma ODi/31$, where: ODi is the absorbance value from each well minus the value of the control sample ($H_2O$) for each plate. The rhizosphere soils from the wheat varieties collected in the summer were characterized by higher AWCD values compared to the soils collected in spring (Figure 6). The highest AWCD value was found in the samples collected in the summer (2018) under the cultivars of Estivus, Pokusa and Jantarka. The lowest AWCD value was found in soil samples collected from the variety of Markiza and KWS Ozon.

On the first season of soil sampling (May 2017) the highest biological activity after 144h incubation of Biolog EcoPlates was characterized by soils from the cultivation of wheat varieties Ostka Strzelecka (Figure 7). The lowest biological activity was found in the soils from the varieties of Fidelius, KWS Ozon and Estivus (Figure 7).

On the basis of cluster analysis (66% Sneath's criteria), three main groups were isolated (Figure 8). The first group includes the following varieties: Linus, Jantarka, Ostka Strzelecka, Pokusa and Ozon. The second group includes the Markiza and Fidelius varieties. In the third group, the following varieties were determined: Rokosz, KWS Ozon, Estivus, Hondia and Belissa (Figure 8).

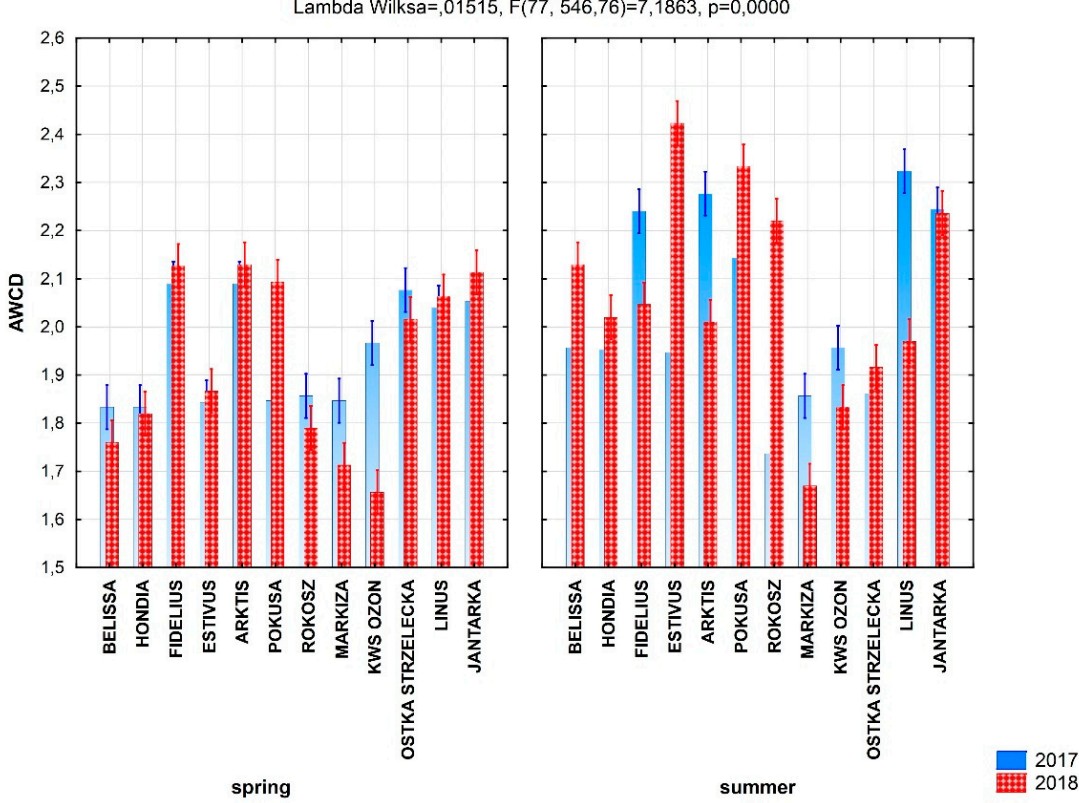

**Figure 6.** The average well-color development (AWCD)—three-way analysis of variance.

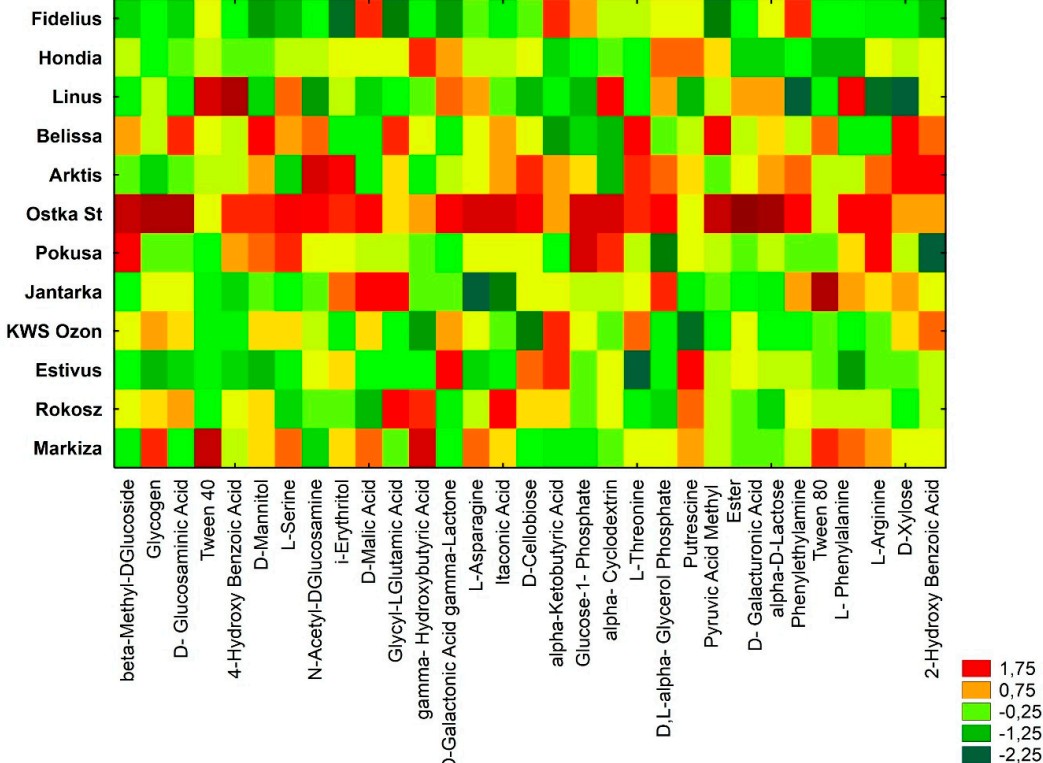

**Figure 7.** Microbial diversity of rhizosphere microorganisms in soil collected on spring 2017 from various winter wheat varieties cultivated in organic farming, (*n* = 3). HeatMaps for the carbon utilization patterns of the 31 substrates located on the Biolog EcoPlates data incubated for 144 h from soil samples.

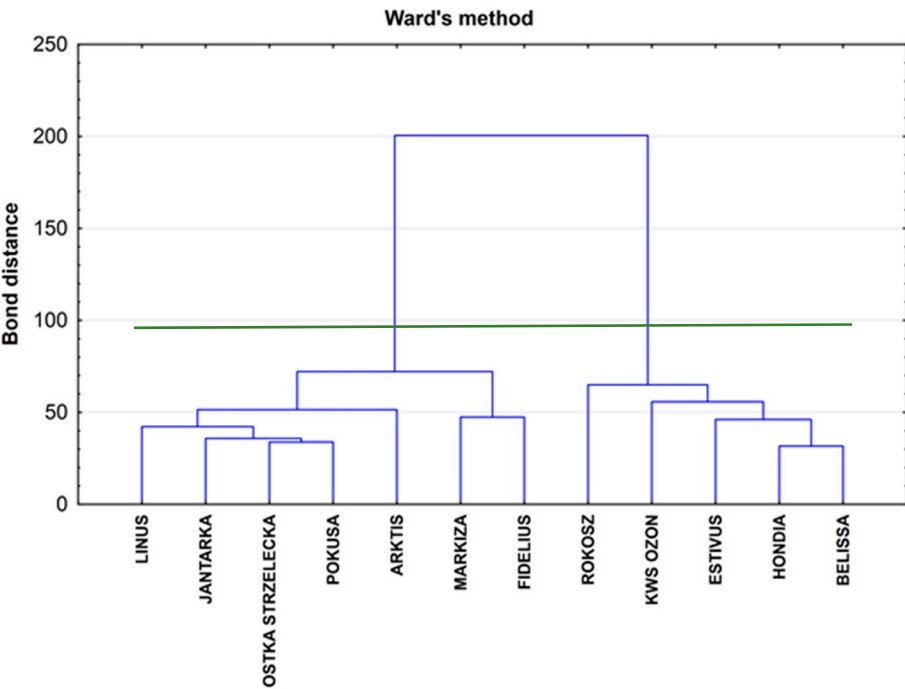

**Figure 8.** Ward's diagram for the carbon utilization patterns of the 31 substrates located on the Biolog EcoPlates data incubated for 144 h from soil samples (spring 2017).

From the 31 different carbon sources, the highest rates of substrate utilization were recorded for the spring rhizosphere soil collected from winter wheat varieties in 2018 such as Arktis, Fidelius, Ostka Strzelecka, Jantarka, Pokusa and Linus (Figure 9). The D-Gluconic Acid gamma-Lactone, Glucose-1-Phosphate, Putrescine, L-Threonine, Glycyl-ʟ-Glutamic Acid, Glycerol Phosphate, L-Serine, L-Asparagine, and D-Malic Acid were the most commonly utilized substrates (Figure 9). The current study revealed that carbohydrates and amines/amides were characterized by the highest metabolic activity, while the lowest activity was determined for polymers. Cluster analysis, including grouping of the treatments and features, was performed on standardized data from the average absorbance values at 144 h (Biolog EcoPlate). The dendrogram was prepared with scaled bond distances on the axis (Ward's method) and boundary marked according to 66% Sneath's criteria. Based on Ward's cluster analysis, two main groups were distinguished for samples collected in spring (Figure 10). The first group included winter wheat varieties such as Linus, Jantarka, Ostka Strzelecka, Pokusa, Arktis, Markiza and Fidelius, while the second group included Rokosz, KWS Ozon, Estivus, Hondia and Bellisa (Figure 10).

In the second term of soil sampling (July 2017, summer) the highest biological activity after 144h incubation of Biolog EcoPlates was characterized by soils from the cultivation of wheat varieties Hondia, Pokusa, Jantarka, and Estivus. The lowest biological activity was found in the soils from cultivating Belissa and KWS Ozon (Figure 11).

Based on Ward's cluster analysis, three main groups were distinguished for samples collected in summer 2017 (Figure 12). The first group included (66%) such winter wheat varieties such as KWS Ozon, Markiza and Fidelis; the second group with the highest activity included such varieties as Rokosz and Arktis. The third group included such varieties as Estivus, Linus, Jantarka, Pokusa, Ostka Strzelecka and Bellisa (Figure 12).

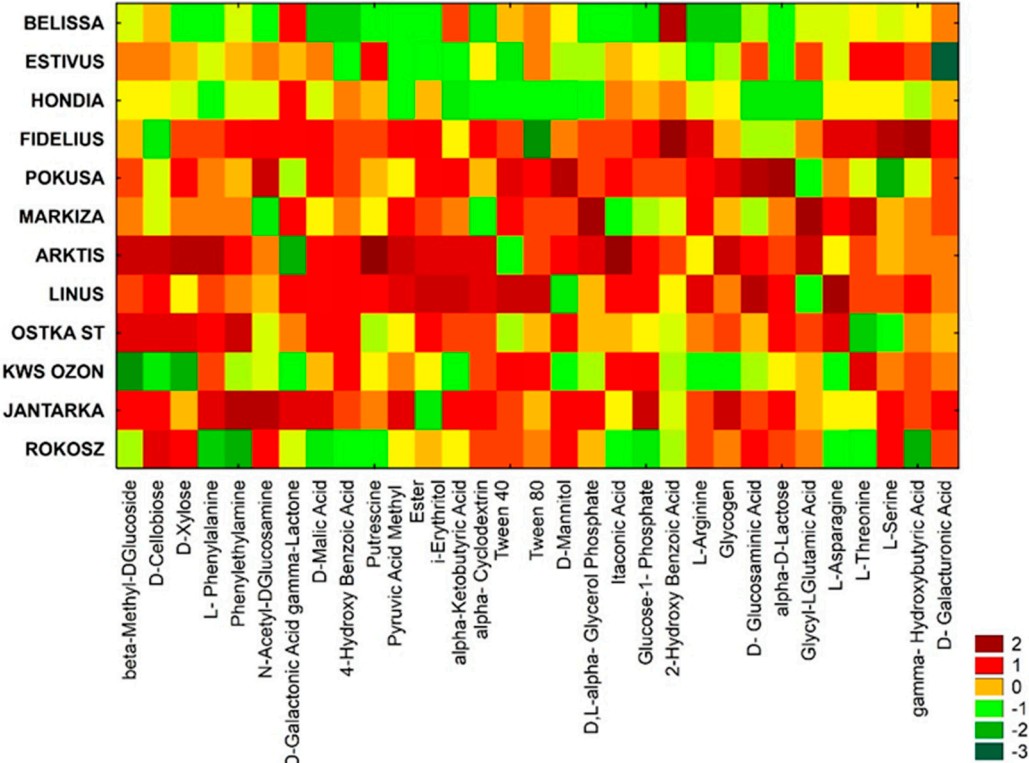

**Figure 9.** Microbial diversity of rhizosphere microorganisms in soil collected on spring 2018 from various winter wheat varieties cultivated in organic farming (*n* = 3). HeatMaps for the carbon utilization patterns of the 31 substrates located on the Biolog EcoPlates data incubated for 144 h from soil samples.

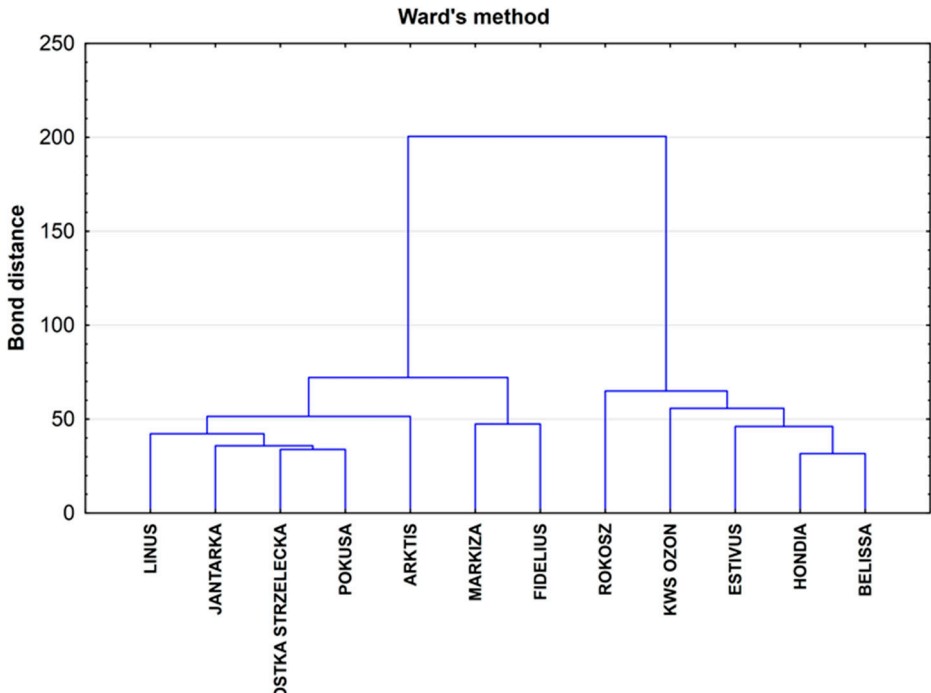

**Figure 10.** Ward's diagram for the carbon utilization patterns of the 31 substrates located on the Biolog EcoPlates data incubated for 144 h from soil samples; spring 2018, *n* = 3.

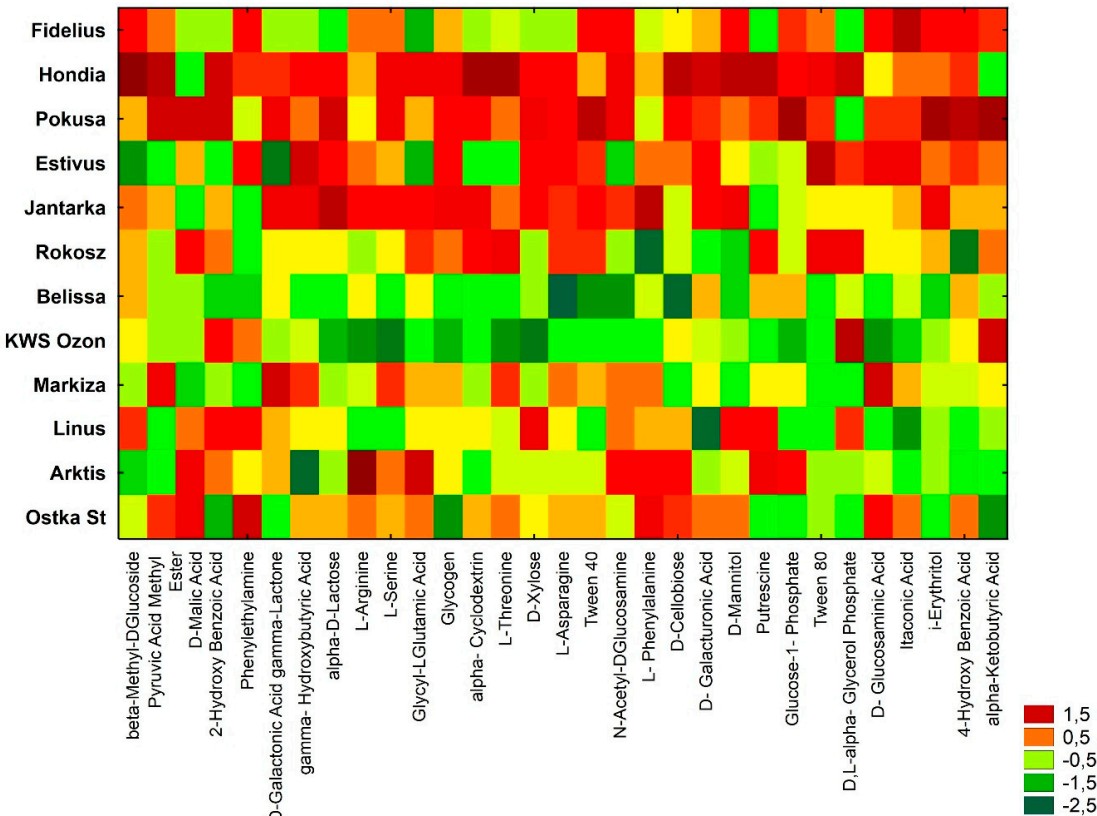

**Figure 11.** Microbial diversity of rhizosphere microorganisms in soil collected on summer 2017 from various winter wheat varieties cultivated in organic farming (*n* = 3). HeatMaps for the carbon utilization patterns of the 31 substrates located on the Biolog EcoPlates data incubated for 144 h from soil samples.

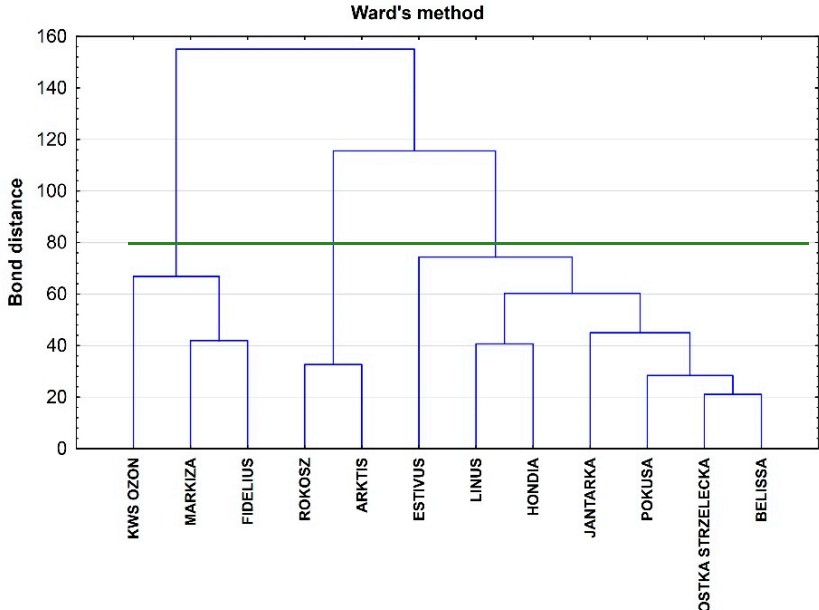

**Figure 12.** Ward's diagram for the carbon utilization patterns of the 31 substrates located on the Biolog EcoPlates data incubated for 144 h from soil samples; summer 2017, *n* = 3.

In summer 2018 the higher activity in substrate utilization was found. The highest rates of substrate utilisation were recorded for rhizosphere soil collected from winter wheat varieties such as Bellisa, Arktis, Rokosz, Estivus, Jantarka and Ostka Strzelecka (Figure 13).

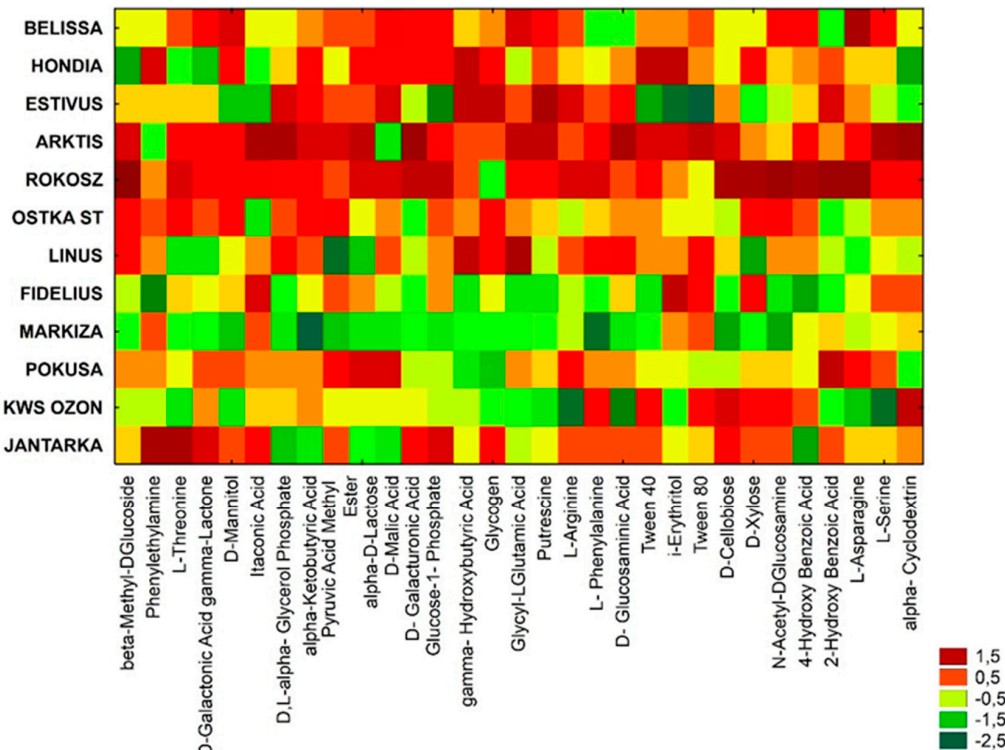

**Figure 13.** Microbial diversity of rhizosphere microorganisms in soil collected on summer 2018 from various winter wheat varieties cultivated in organic farming (*n* = 3). HeatMaps for the carbon utilization patterns of the 31 substrates located on the Biolog EcoPlates data incubated for 144 h from soil samples.

Based on Ward's cluster analysis, three main groups were distinguished for samples collected in summer (Figure 14). The first group included such winter wheat varieties such as KWS Ozon, Markiza and Fidelis; the second group with the highest activity included such varieties as Rokosz and Arktis. The third group included such varieties as Estivus, Linus, Jantarka, Pokusa, Ostka Strzelecka and Bellisa.

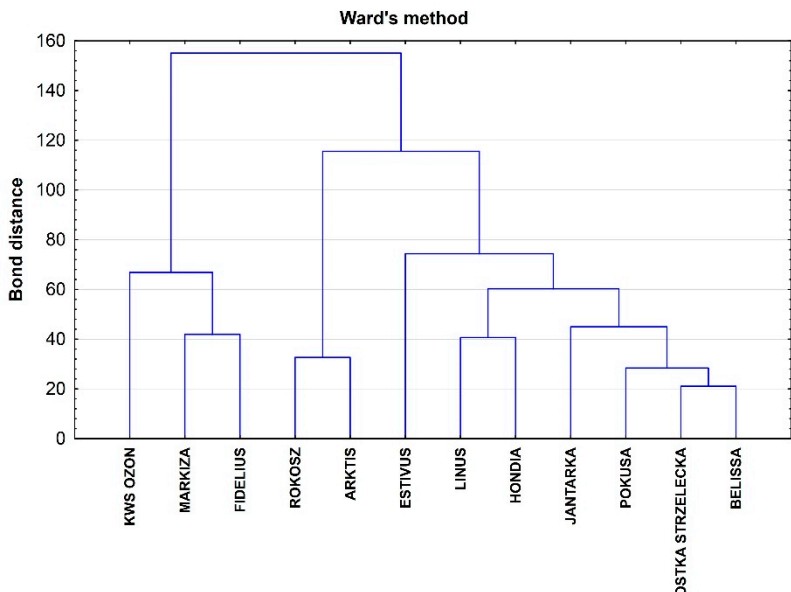

**Figure 14.** Ward's diagram for the carbon utilization patterns of the 31 substrates located on the Biolog EcoPlates data incubated for 144 h from soil samples; summer 2018.

Based on the PC (principal component) analysis in samples collected in spring 2017 the three groups of the following varieties of winter wheat were distinguished on the basis of the metabolic profile:

-      Ostka Strzelecka (with very high biological activity);
-      Arktis, Estivus, Jantarka, Belissa, Rokosz, Hondia, Fidelius, KWS Ozon, Marquise (with high and average biological activity);
-      Pokusa, Linus (with the lowest biological activity) (Figure 15).

In the same year (2017) in the samples taken in the summer, the metabolic profile of individual winter wheat varieties was grouped as follows:

-      I group: Rokosz, Estivus, Markiza, KWS Ozon, Hondia and Belissa;
-      II group: Arktis, Ostka Strzelecka, Jantarka, Pokusa and Linus;
-      III group: Fidelius (Figure 16).

Based on the analysis of the main components in soil samples collected in summer 2017, three groups focusing on the following types of winter wheat were distinguished on the basis of the metabolic profile of soil rhizosphere:

-      Ostka Strzelecka, Belissa and Markiza (with high soil biological activity);
-      Estivus, Rokosz, Arktis, Fidelius, Linus and KWS Ozon (with low soil biological activity);
-      Hondia, Pokusa, Jantarka (with low biological activity) (Figure 17).

The highest biological activity was characterized by soils collected in summer 2018 from winter wheat varieties such as Arktis, Hondia Pokusa, Rokosz and Ostka Strzelecka (Figure 18).

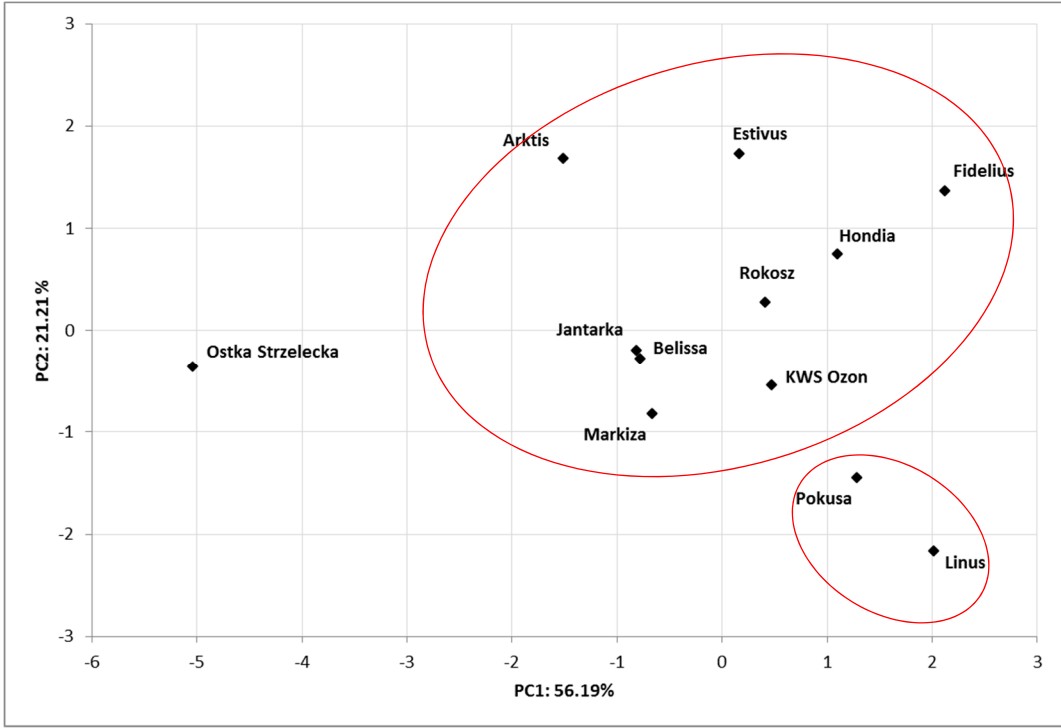

**Figure 15.** Principal component analysis of biodiversity indexes of Biolog EcoPlates incubated for 144 h from soil samples and various winter wheat varieties cultivated in organic farming, spring 2017 (*n* = 3).

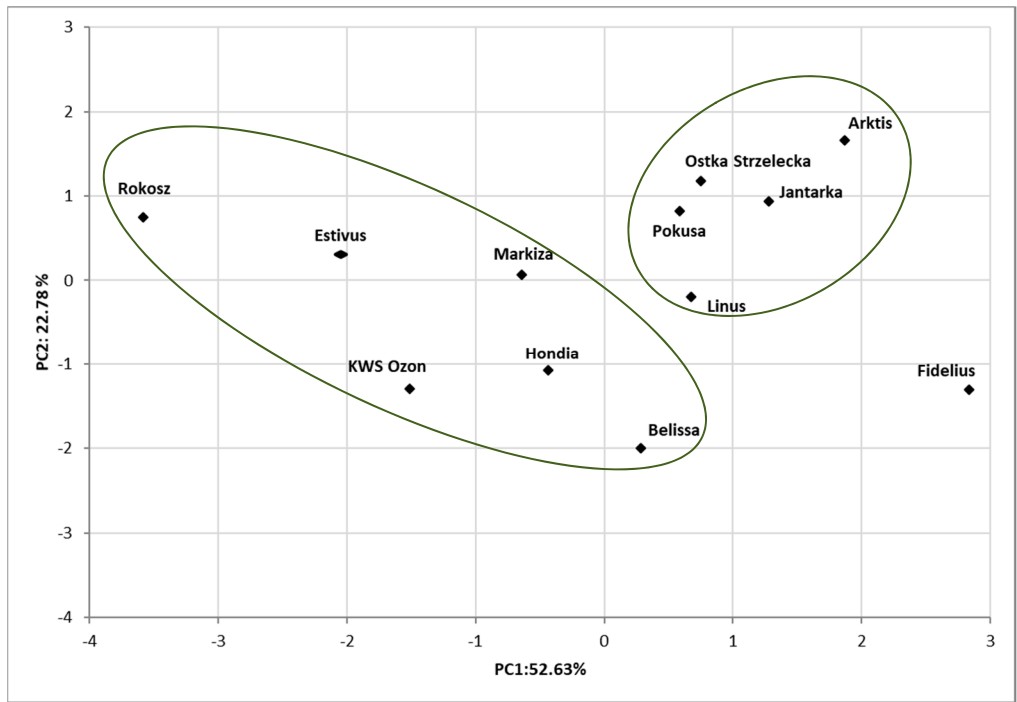

**Figure 16.** Principal component analysis of biodiversity indexes of Biolog EcoPlates incubated for 144 h from soil samples and various winter wheat varieties cultivated in organic farming, spring 2018 (*n* = 3).

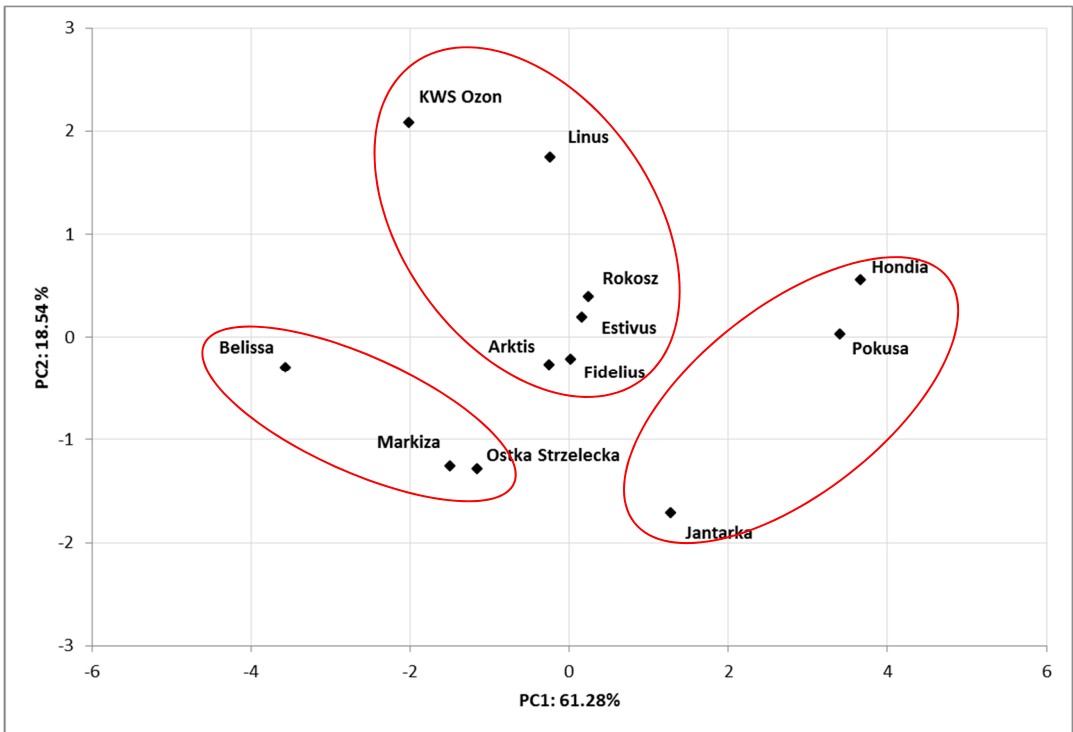

**Figure 17.** Principal component analysis of biodiversity indexes of Biolog EcoPlates incubated for 144 h from soil samples and various winter wheat varieties cultivated in organic farming, summer 2017 (*n* = 3).

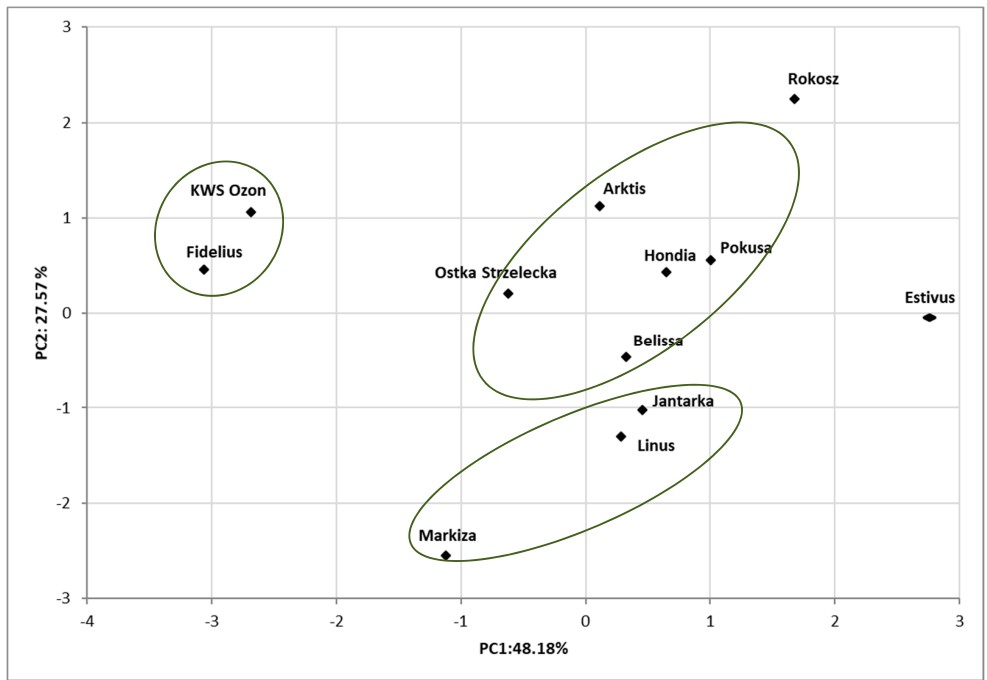

**Figure 18.** Principal component analysis of biodiversity indexes of Biolog EcoPlates incubated for 144 h from soil samples and various winter wheat varieties cultivated in organic farming, spring; average for years 2017–2018 (*n* = 6).

*3.4. The Yield of Plants*

The yield of plants was presented in Table 3. Based on data obtained from 2017–2018, it was found that the highest yield was found for winter wheat varieties such as Bellisa, Fidelius, Hondia and Jantarka. These results were also confirmed in the assessment of the biological activity of rhizosphere soil.

**Table 3.** Yield of plants.

| Year | 2017 | | | 2018 | | |
|---|---|---|---|---|---|---|
| Variety | Yield (t/ha) | Number of Ears (quantity/m$^2$) | The weight of 1000 Grains (g) | Yield (t/ha) | Number of Ears (quantity/m$^2$) | The Weight of 1000 Grains (g) |
| Arktis | 3.12 | 346 | 36.7 | 2.99 | 325 | 33.2 |
| Bellisa | 4.08 | 324 | 44.2 | 3.37 | 314 | 35.7 |
| Estivus | 3.51 | 323 | 41.6 | 3.72 | 321 | 38.5 |
| Fidelius | 4.17 | 326 | 42.4 | 3.92 | 335 | 37.7 |
| Hondia | 3.97 | 338 | 44.9 | 3.89 | 303 | 44.7 |
| Jantarka | 3.95 | 356 | 43.5 | 3.97 | 333 | 42.3 |
| KWS Ozon | 3.00 | 341 | 40.8 | 3.30 | 308 | 38.7 |
| Linus | 3.64 | 288 | 37.6 | 2.78 | 312 | 38.6 |
| Markiza | 3.60 | 326 | 39.8 | 3.62 | 352 | 34.7 |
| Ostka Strzelecka | 3.22 | 312 | 40.8 | 3.20 | 299 | 35.1 |
| Pokusa | 3.38 | 293 | 38.2 | 2.87 | 329 | 33.9 |
| Rokosz | 1.89 | 290 | 42.5 | 3.24 | 322 | 35.8 |
| Mean | 3.46 | 322 | 41.1 | 3.37 | 379 | 35.7 |
| *NIR$_{0.05}$* | *0.33* | *93* | *0.95* | *0.49* | *56* | *1.57* |

## 4. Discussion

In contemporary agriculture, modern farming has to meet many requirements promoting soil protection and soil quality enhancement [26,27]. In organic farming the winter wheat varieties are the most popular cultivation crops. However, both for environmental and economic reasons not all of them are useful for organic farming [28]. A very important issue is limitation of soil organic matter losses as well as improvement of soil structure and reduction of its tendency towards encrustation [14,20]. A very important biological factor in the aspect of these problems is the selection of new quality varieties of winter wheat cultivation in organic farming based on evaluation of the microbiological profile of their soil rhizosphere. The interaction between the root (rhizosphere zone) and the other soil organisms (bacteria, fungi) is a very important aspect of the agroecosystem [18,29]. The interaction between microorganisms and rhizosphere is determined by genetic factors; nowadays, this factor is not taken into consideration in the breeding process. Therefore small-plot trials with particular selected winter wheats varieties and determination of biological activities of soil are very important to selection of ne winter wheat varieties.

The ecological crop production (organic farming) is the cultivation using a limited quantity of nutrients and serial applications of nitrogen [30,31]. The winter cereals they constitute this group of plants which have problems with absorbing available nitrogen from the soil mainly in early spring. The ability to absorb nitrogen in early spring is an important issue in the growing of winter varieties of cereals in an organic farming system. Soil mainly in early spring is characterized by low or very low biological activity [32]. Therefore, cereal varieties are sought after which quickly develop the root system and establish active symbiosis with soil microorganisms. A well-developed root system of winter wheats, responsive to the interaction with the soil microorganisms in a positive way, is a very important aspect of the efficient absorption of nutrients and especially nitrogen fixing [9,33]. A long and deep root system with a lot of small young roots forms a better and richer branching with more capillary roots. The deep roots assure the sufficient absorption of water and nutrients from the deeper layers of the soil profile.

The growth of the roots system is more important than the growth of the upper part of plants in a soil characterized by a lower concentration of accessible nitrogen. However, this problem is much more complicated because a well-developed root system of plants is influenced not only by the soil structure, but also by the proportion of nutrients and water in the soil and by the genotype of plants [7]. This means that the selection of suitable varieties for the farming system should be applied to the conditions of organic farming. The selection of an efficient root system, adapted to the absorption of nutrients (especially nitrogen) from the soil, should take into account the limited competitiveness of varieties for the assimilates of the roots [6,34]. Some authors indicating that phosphate solubilizing microorganisms and their activity can play an important role in the performance of winter wheat cultivars under organic farming [35].

In previous studies on rhizosphere microorganisms, the focus was on determining their number and identification of the most important species. Currently it is also possible to analyze the metabolic activity of the general population of microorganisms inhabiting various environments, including the rhizosphere of plants [9]. One such method is the Biolog System [36]. Direct incubation of environmental samples in Biolog EcoPlates with various carbon substrates is used in studies of changes in complex microbial communities and assessment of their metabolic profile. The assessment of the soil community level physiological profiles (CLPP) is one of the methods used in the study of functional diversity of microbial communities under the influence of the abiotic and biotic factors, in this case in the assessment of various winter wheat varieties [37,38]. Research carried out with the use of this system provides valuable information on the functional diversity of microbial communities and can be used to monitor environmental changes in the context of assessing the activity of microorganisms in the rhizosphere of different varieties of winter wheat cultivated in the ecological system [39].

The investigations of the impact of plant production systems and soil cultivation on the microbiological quality of the soil environment indicated by the selected parameters of soil biological

quality allowed a profound analysis of not only quantitative but also qualitative changes occurring in soil organic matter and microbial communities. They also allowed assessment of these relationships in a broader aspect of soil productivity protection. The microbiological diversity can be limited in natural conditions by undesirable environmental factors, e.g., limited nutrient resources, ecological and physical factors exceeding the tolerance of the organism, cultivation systems, and interspecies interactions preventing the occurrence or sustenance of a species in the environment. The selection of cereal varieties suitable for a given soil can increase the biodiversity and biological activity of soils.

Microorganisms inhabiting plant roots indirectly influenced the formation of biological activity and soil fertility—which is particularly important in the case of organic farming. High variability of biological activity of rhizosphere soils in the growing season between particular varieties of winter wheat was observed. The results of these analyses allowed for a more complete characterization of the yield potential of the tested varieties and their suitability for cultivation in the conditions of organic farming.

## 5. Conclusions

Microbial diversity in rhizosphere soils of new quality varieties of winter wheat cultivation in organic farming was dependent on the variety as well as the year and the sampling season.

1.  The highest dehydrogenase activity was observed in the rhizosphere soil of the following wheat varieties: Fidelius, Estiwus, Arktis, Pokusa, Rokosz collected in spring, and Rokosz for the second phase of sampling, i.e., summer.
2.  The highest activity of alkaline phosphatase was found in the rhizosphere soil of the Estivus Pokusa, Rokosz, Markiza, Ostka Strzelecka varieties in soil collected from rhizosphere soil in summer. In the soils collected in spring, the highest activity of acid phosphatase was characterized by soil from wheat varieties Belissa, Estivus and Arktis.
3.  The highest alkaline phosphatase activity was found in the rhizosphere soil collected in spring of KWS Ozon cultivar. In soil collected from rhizosphere soil in summer the higher alkaline phosphatase activity was found in the rhizosphere soil of such varieties as Hondia, KWS Ozon and Jantarka.
4.  The highest AWCD value was found in the samples collected in the summer under the cultivars of Estivus, Pokusa and Jantarka. The lowest AWCD value was found in soil samples collected from the varieties of Markiza and KWS Ozon.
5.  Some varieties of winter wheat such as Pokusa, Rokosz, Markiza, KWS Ozon and Ostka Strzelecka can be cultivated in organic farming while maintaining high biological activity of soils.

**Author Contributions:** Conceptualization, A.G.; methodology, A.G. and E.G.; software, A.G.; validation, A.G.; formal analysis, A.G. and E.G.; investigation, A.G. and E.G. and K.J.; resources, A.G.; data curation, A.G. and K.J.; writing—original draft preparation, A.G.; writing—review and editing, E.G. and K.J.; visualization, A.G.; supervision, A.G.; project administration, A.G. and K.J.; funding acquisition, A.G. and K.J.

**Funding:** The research was financed from the research subject under subsidies for organic farming. Ecological Variety Experimentation, assumptions and implementation of the system. Decision of the Minister of Agriculture and Rural Development No.: HOR.re.027.6.2018 (1).

**Conflicts of Interest:** The authors declare no conflict of interest.

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
