# Peer review of "Changes of Microbial Diversity in Rhizosphere Soils of New Quality Varieties of Winter Wheat Cultivation in Organic Farming"

_sustainability, doi:10.3390/su11154057_

Round 1
Reviewer 1 Report
Let me propose the following small modifications:
line 13: delete "of"
line 28: allows
line 84: to evaluate the changes OR evaluation of changes
line 86: relevant to
line 87: farming
line 100: sentence to the past tense (in my opinion...)
line 108: in spring
line 140: conditions during
line 151: seasons
line 155: "respectively" et the end of sentence
line 158: wheats
line 177: different
line 211-212: ?are the values correct (according the table 3)?
line 219: in spring
line 243: in summer
line 249: "such"-only once in the sentence
line 309: they constitute this the group
line 310: have has
line 320: roots system
line 327: indicate indicating

Author Response
Dear Revisers
Thank you for your valuable comments regarding the publication. All comments are included in the manuscript and the amendments are in the text (red).
Concerning detailed comments:
1. in accordance with the reservations of Reviewer 3 regarding calculations, all results have been included separately in two years. New calculations have been made. The distribution was normal. Hence, the manuscript has expanded considerably by a larger number of charts and tables.
2. Each cultivar was sown (4.5 million grains/ha) in 3 replicated and randomized micro-plots (35 m2). From each micro-plots (layer of 0-20 cm) were taken several subsamples and combined from one (biological replicate). In this way, 3 biological repetitions were obtained. In addition, each sample was made in three technical repetitions.
3. language correction was made.

Reviewer 2 Report
Broad comments:
The strength of the manuscript is in an interestingly chosen topic whose results can improve organic farming in the field of plant production, especially cereals. The manuscript is conceived as a result of analyzes that would allow a more complete characterization of yield potential of winter wheat varieties tested and their suitability for cultivation in organic farming.
Weaknesses in the manuscripts are in improper interpretation of the obtained results, poor knowledge of statistical data processing and inconsistencies in the presented data. Details of the methods and statistical analysis were not sufficiently provided to evaluate the manuscript. The authors seemed to rush through the methods and the presentation of results without providing the details needed to properly evaluate. In the section Discussion, the authors did not discuss the obtained results or interpreted them in the perspective of previous studies and working hypotheses. Conclusions are general and do not come from research results.
Specific comments
L87 instead forming – farming
L119-120 methodology should be briefly described
L135-138 what about correlation? There are mentioned correlations in the manuscript
L147 Climatic conditions during the vegetation period are not well explained (neither precipitation nor temperature). Authors did not highlight significant differences the precipitation and temperature. It seems to me that the difference of 35 mm (in April) is not small and these values are not similar. The same situation is with temperature - difference is 6 °C. The recorded results are not related to long term period.
L154-155 says that the highest total bacteria number was recorded at three wheat varieties, but differences in total bacteria number between varieties Pokusa and KWS Ozon, and Pokusa and Linus are statistically significant while the difference between KWS Ozon and Linus are not.
L157-164 incorrectly interpreted results. Follow the letter markings in the table that signify a statistically significant difference
L171-173 in the Table 2 didn`t shows the statistical difference between the parameters of biological activity collected in spring and summer
L176-179 Table 2. instead bacteria - B, instead Fungi – F. All abbreviation should stand in the table not just some of them
L193 not only Estivus variety. There are no significant differences between AcP of Estivus var. and Arktis, Balissa, Hondia. It is necessary to properly and accurately explain the statistics for DHA, AcP and AIP in spring and summer time of soil collection
L200-201 the table does not have a list of character labels
L211- says Shanon index was 3.50 – in Table 3 this value is 3.60. Match numbers in text and table.
L212 – the same situation like as previous statement – in text – 3.34 – in table – 3.47
L252 Hondia did not included. Why?
L259 what is PC1? PC2?
L288-293 Why the yield was not statistically processed? Why yields and yield components are not mentioned in materials and methods?
L291-292 explain the statement
L294-342 Discussion is not based on the results but has been written generally. It is necessary to completely change the discussion with a compulsory interpretation of the obtained results and link the obtained results with some the previous research other authors.
L346 what soil cultivation? No soil cultivation treatment is specified in manuscript.
L346-361 Conclusions are general and do not come from research results
Author Response

(The authors gave the same response as above.)

Reviewer 3 Report
This is a potentially interesting report addressing the effect of wheat varieties on rhizospheric microorganisms. I have some points that deserve the author’s attention and need to be clarified before a decision could be made. In the materials and methods authors cite that” Each cultivar was sown (4.5 million 99 grains ha-1) in 4 replicated and randomized micro-plots “ Therefore we expect data to be the means of four replicates. However, in Table 2 the legend indicates: “ ….average for years 2017-2018 (n=6).”
If there were 4 replicates and two years are averaged one would expect that number to be n=8. Why are two plots missing ?
In second instance, the authors thereby declare to have averaged the data of the two different years. This is not a starightforward nor legitimate choice since those data are not real replicates being collected in two years where soils would have logically encountered not identical conditions and communities would have in any event undergone an evolution. Replicates need to be taken in parallel and not in differnt times. Moreover if they sampled in the same plots in different times there would even be an issue of self correlation between pairs of samples.
The authors need to:
1) Provide to this reviewer the original datasets featuring the independent data of the two years for the four replicate plots
2) Perform the means of each group of four values by keeping separated the two years (2017 and 2018).
3) Before repeating the statistics on those data they need to verify that the chosen test (ANOVA with Tukey’s post hoc) was actually allowed by the fulfillment of its requirements in terms of data distribution: normality (tests of Shapiro Wilk or equivalent) and homoscedasticity ( Levene’s test).
4) Perform the corresponding statistics on the separated two years data.
Upon clearing the above, and revising results and discussion if the new analyses should call for it, their work will be reevaluated.
Author Response

(The authors gave the same response as above.)

Round 2
Reviewer 2 Report
-